# A High Throughput Apoptosis Assay using 3D Cultured Cells

**DOI:** 10.3390/molecules24183362

**Published:** 2019-09-16

**Authors:** Sang-Yun Lee, Il Doh, Dong Woo Lee

**Affiliations:** 1Department of Health sciences and Technology, SAIHST, Sungkyunkwan University, Seoul 06351, Korea; leesangyun316@gmail.com; 2Medical & Bio Device (MBD), Suwon 16229, Korea; 3Center for Medical Convergence Metrology, Korea Research Institute of Standards and Science, Daejeon 34113, Korea; 4Department of Biomedical Engineering, Konyang University, Daejon 35365, Korea

**Keywords:** apoptosis assay, caspase-3/7, 3D cell culture, cell encapsulation, high-throughput screening

## Abstract

A high throughput apoptosis assay using 3D cultured cells was developed with a micropillar/microwell chip platform. Live cell apoptosis assays based on fluorescence detection have been useful in high content screening. To check the autofluorescence of drugs, controls (no caspase-3/7 reagent in the assay) for the drugs are necessary which require twice the test space. Thus, a high throughput capability and highly miniaturized format for reducing reagent usage are necessary in live cell apoptosis assays. Especially, the expensive caspase-3/7 reagent should be reduced in a high throughput screening system. To solve this issue, we developed a miniaturized apoptosis assay using micropillar/microwell chips for which we tested seventy drugs (six replicates) per chip and reduced the assay volume to 1 µL. This reduced assay volume can decrease the assay costs compared to the 10–40 µL assay volumes used in 384 well plates. In our experiments, among the seventy drugs, four drugs (Cediranib, Cabozatinib, Panobinostat, and Carfilzomib) induced cell death by apoptosis. Those results were confirmed with western blot assays and proved that the chip platform could be used to identify high potency apoptosis-inducing drugs in 3D cultured cells with alginate.

## 1. Introduction

Three-dimensional (3D) cell culture has been studied as a way to increase the biological relevance of disease models and the predictive value of drug studies [1,2]. These 3D models typically increase the level of cell–cell interactions and reduce or eliminate the dominant cell–substrate interactions seen in conventional two-dimensional (2D) monolayer cultures. When cells derived from cancer patients are cultured in 3D, the cell–cell and cell–extracellular matrix (ECM) interactions change the morphology of the cells and the type and expression level of major genes [3,4,5,6]. For these reasons, 3D cell culture tools have been widely studied, and some of them have already been commercialized. Most of the commercialized 3D cell culture tools are based on a 96-well plate, and some have increased the throughput with a 384-well plate [7,8]. 3D cell cultures are used to evaluate drugs’ efficacy/toxicity or mechanism of action (MOA) by staining 3D cultured cells with various cell-dyeing reagents [9,10]. However, in the case of biomarker staining for determining the mechanism of action (MOA) of the drug, a large amount of expensive dyeing reagent is still required for high content screening (HCS).

To solve this issue, micropillar/microwell chip platforms for a 3D cell-based assay were developed. Previously, we miniaturized a drug efficacy assay using a 3D cell culture with a micropillar/microwell chip [11,12,13,14]. The media or reagents in the microwell chip, which were exposed to the 3D cultured cells on the micropillar chip, could be changed by replacing the microwell chip with one containing fresh media or reagents. This washing method makes it possible to reduce the size of well and pillar and the amount of dyeing reagent for staining the cells. Compared to the 10–40 µL/well of dyeing reagent for a 384-well plate, the micropillar/microwell chip requires only 1 µL/well. 

In this paper, we applied the apoptosis assay to the micropillar/microwell chip shown in Figure 1. In a single chip (38 × 14 micropillar array), we tested 72 drugs (including two controls) in a 36 × 12 array. As shown in Figure 1A, six spots were used for one drug. Three spots were without the caspase-3/7 staining dye, and the other three spots were with the caspase-3/7 staining dye. Examination of the cleaved-caspase-3/7 signal is the general apoptosis assay for the MOA analysis. Normally, western blot assay, as a molecular biological analysis method, is performed to confirm the cleaved-caspase 3/7 protein signal [15,16]. However, traditional western blot analysis requires complex steps to identify the cleaved-caspase 3/7 protein signals expressed in cells. Therefore, it is inefficient to analyze the MOAs of many drugs when performing high-throughput drug screening. To overcome these problems and to quantitatively measure the activated caspase 3/7 protein signals, we applied the apoptosis assay to image-based high-throughput drug screening. Especially, detection of activated caspase-3/7 from the live cells in the micropillar/microwell chip could be simplified by eliminating cell fixation. We used CellEvent^®^ Caspase-3/7 Green (Thermo Fisher Scientific) for the detection of activated caspase-3/7. CellEvent^®^ Caspase-3/7 Green reagent is a four amino acid peptide (DEVD) conjugated to a nucleic acid-binding dye that is nonfluorescent when not bound to DNA because the DEVD peptide inhibits binding of the dye to DNA. When cleaved caspase-3/7 is expressed in apoptotic cells, the DEVD peptide is cleaved, and the free dye can bind DNA, generating a green fluorescence [17]. However, the autofluorescence of drugs must be checked, which requires additional space for a control test. The miniaturized micropillar/microwell chips, containing 532 test spaces in a 75 mm by 25 mm area, could provide the additional capacity for the control tests. As a proof of concept, an apoptosis test of seventy drugs was conducted in one chip.

## 2. Results and Discussion

### 2.1. Autofluorescence Detection

Some cells or drugs emit natural light, and this could overlap with the wavelength range of the dyeing reagent. In the immunostaining, the cells were fixed, and a lot of washing steps removed residual drugs, and a quenching step was also done to reduce the background or autofluorescence. Autofluorescence is a serious problem in color-based analysis in live cells. Drugs inside the cells did not wash out, and the quenching step in the immunostaining was not available for the live cells. Four drugs (Bosutinib, PHA-665752, Sunitinib Malate, and Sotrastaurin) from among the seventy drugs showed green fluorescence in the unstained cells shown in Figure 2A. The green area of the stained and unstained cells with caspase-3/7 looked similar, and the *p*-value between the two groups is higher than 0.05, which means the two groups are not significantly different (Figure 2A). By comparing staining and unstaining tests for all drugs, we excluded activated caspase-3/7 because of the drugs’ fluorescence.

### 2.2. Multiparameter Analysis for the Apoptosis Assay

As shown in Figure 2B, the green area was detected in the stained cells but not in the unstained cells. A *p*-value less than 0.01 between two groups shows that activated caspase-3/7 was significantly overexpressed, and the apoptosis rate was higher (Figure 2B). Before calculating the apoptosis rate, the *p*-value between the stained and unstained cells were calculated, and any *p*-values at 0.05 or higher were excluded. Apoptosis is programmed cell death that involves the controlled dismantling of intracellular components while avoiding inflammation and damage to surrounding cells. Generally, apoptosis induction may occur easily when 3D cultured cells are exposed to drugs. After caspase-3/7 was activated, it took time for apoptosis and cell death to occur. Thus, we stained caspase-3/7 at day 2 (one day of drug treatment) and compared those results with the viabilities at day 7 (6 day drug treatment) shown in Figure 4. In the experiment, four drugs showed highly activated caspase-3/7, and all drugs killed 60% or more of the 3D cells. The seven-day cell viabilities of those drugs were lower than 40%, as shown in Figure 3. Figure 4A shows the seven-day viabilities and two-day apoptosis rates according to the seventy drugs. Four drugs (Cediranib, Cabozatinib, Panobinostat, Carfilzomib) showed a high apoptosis rate. To verify the chip data, the activated caspase-3/7 levels of the 3D cells exposed to the four drugs were measured by western blot. However, we have not tested all 71 drugs because it was too many to do western plots for. Thus, four drugs that strongly induced caspase 3/7 were tested. As shown in Figure 4C, the four drugs highly exhibited activated caspase-3 (more than 40%) which was correlated with the measurement of the chips. Thus, we only proved that the chip platform was intended to identify high potency apoptosis-inducing drugs in 3D cultured cells. The highly apoptosis-inducing drugs were already known in previous reports [18,19,20,21]. Cediranib (#30) is a protein tyrosine kinase inhibitor that potently inhibits VEGF receptor-2. It was found to potently inhibit VEGF-A-induced VEGFR-2 phosphorylation and can regulate cell proliferation, apoptosis, invasion, and migration in human umbilical vein endothelial cells and some tumor cells [18]. Cabozantinib (#37) is an oral multikinase inhibitor. The principal targets are receptor tyrosine kinases of cancer cell growth and tumor angiogenesis including MET, RET, AXL, and VEGFR2 [19]. Cabozantinib showed a significant increase in cleaved caspase 3 in sensitive colorectal cancer. Panobinostat (#47) induces apoptosis by production of reactive oxygen species and synergizes with topoisomerase inhibitors in cervical cancer cells [20]. Carfilzomib (#60) induces cell cycle arrest and apoptosis and potentiates the antitumor activity of chemotherapy in rituximab-resistant lymphoma [21].

## 3. Materials and Methods

### 3.1. Experimental Procedure

The micropillar chip was made of poly(styrene-co-maleic anhydride) (PS-MA) and contained 532 micropillars (with a 0.75 mm pillar diameter and a 1.5 mm pillar-to-pillar distance) [12]. PS-MA provides a reactive functionality to covalently attach poly-L-lysine (PLL), ultimately attaching alginate spots by their ionic interactions as shown in Figure 1A. Thus, cells in alginate spots on the micropillar did not detach during culturing and changing media. As shown in Figure 1B. cells grew and formed spheroids in alginate spots compared to the flat cell morphology in 2D cell culture. Plastic molding was performed with an injection molder (Sodic Plustech Inc., Schaumburg, IL, USA). Approximately 100 cells (A549 cell line) in 50 nL with 0.5% (*w*/*w*) alginate were automatically dispensed onto a micropillar chip with the ASFA™ Spotter ST (Medical & Bio Decision, South Korea). The ASFA™ Spotter ST uses a solenoid valve for dispensing the 50 nL droplets of the cell–alginate mixture and the 1 µL of the media, drugs, or dye reagent. After dispensing the cells and media into the micropillar and microwell, respectively (Figure 5a), the micropillar chip containing the cells in alginate was combined (or “stamped”) with the microwell chip filled with the 1 µL of the fresh media (Figure 5b). The combined chips were incubated for one day at 37 °C and 5% CO_2_ to stabilize the cells. Then, the micropillar chips were moved to a new microwell chip filed with the drugs for the drug treatment. Two sets of chips were made for apoptosis and viability assays, respectively. Two days after the drug treatment, one of the two microwell chips was prepared with HOECHST (Blue), caspase-3/7(Green), and Calcein AM(Red), as shown in Figure 5c. This microwell chip was combined with the micropillar chip containing the cells for 6 h at room temperature for the staining (Figure 5d). After the staining, micropillar chips were quickly dried at room temperature. The 50 nL alginate spot containing the cells quickly became flat. The cells in the dried alginate spot were easily and quickly visualized with an automatic microscope with a three-color filter without deconvolution technology. (Figure 5e). Six days after the drug treatment, the 3D cultured cells of the other micropillar chip were stained with Calcein AM for the cell viability assay.

### 3.2. Cell Culture

Human lung carcinoma, A549, was purchased from the Korean Cell Line Bank (Seoul, Korea). A549 was cultured in T-75 cell culture flasks (Eppendorf, Vienna, Austria) filled with RPMI 1640 medium (Gibco, Co Dublin, Ireland) supplemented with 10% fetal bovine serum (FBS, Gibco, Co Dublin, Ireland). Cell lines were cultured at 37 °C in a 5% CO_2_-humidified cell incubator (Sheldon Manufacturing, Cornelius, OR, USA) and passaged every four days at 60% confluence. For the experiment, two-dimensionally incubated cells were treated with 1X TrypLE Express enzyme (Gibco, Co Dublin, Ireland) to separate from the T-75 flask bottom. Then, the cells were collected in a 50 mL falcon tube. After centrifugation at 2000 rpm for 3 min., the supernatant was removed, and the cells were resuspended with RPMI 1640 conditioned media to a final concentration of 10 × 10^6^ cells/mL. The number of cells in the RPMI 1640 conditioned media was calculated with the AccuChip cell counting kit (Digital Bio Inc., Seoul, South Korea). The rest of the cells were seeded at a concentration of 1 × 10^6^ cells in a T-75 flask containing 15 mL of RPMI 1640 conditioned media. Additionally, we used the A549 cell line under 20 passages after thawing from a frozen cell stock. Under 20 passages, we observed that the A549 cell line easily formed 3D cells in 0.75% (*w*/*w*) alginate on the chip platform.

### 3.3. Western Blot Assay

After treatment with the four drugs (Cediranib, Cabozatinib, Panobinostat, and Carfilzomib) induced apoptosis, 3D cells were recaptured, and then 100 µL of cOmplete™ Lysis-M buffer solution (Roche Life Science, Germany) was added. Cell lysates were vortexed for 30 sec., and then the sample was placed at 4 °C for 30 min. After centrifuging at 15,000 rpm for 5 min., protein concentrations were measured using the BCA protein assay (Pierce, Rockford, IL, USA). Protein extracts were resolved using 4–20% Mini-PROTEAN TGX™ Precast Protein Gels (Bio-Rad, Hercules, CA, USA) and transferred onto iBlot^®^ PVDF gel Transfer Stack membranes (Thermofisher Scientific, Korea). After blocking nonspecific binding sites for 1 h in 5% bovine serum albumin (BSA) in Tris-buffered saline containing 0.1% Tween-20 (TBS-T), the membranes were incubated overnight at 4 °C with specific primary antibodies. The antibodies were Cleaved Caspase-3 (Asp175) antibody (1:1000, Cell Signaling Technology, Inc., Boston, MA, USA) and anti-beta actin (1:2000, Abcam, Cambridge, MA, USA). These were used in accordance with the manufacturers’ instructions.

### 3.4. Multicolor Live Cell Staining and Scanning

In the live cell staining, the staining buffer contained 5 mL of 140 mM NaCl with 20 mM CaCl_2_. CaCl_2_ was used to prevent degradation of the alginate spots. A staining dye solution was prepared by adding 1 µL of calcein AM (4 mM stock from Invitrogen), 1 µL Hoechst 33342, and 10 µL CellEvent^®^ Caspase-3/7 into 8 mL of staining buffer. The staining dye solution was dispensed at 1 µL per microwell in the microwell chips which was then combined with the micropillar chip containing the cells for 6 h at room temperature. The 3D cultured cells required more than 6 h to measure the activated caspase-3/7 while 30 min or 1 h was enough in the 2D cultured cells. After staining, the cells were washing twice with the staining buffer for 30 min. The alginate spot containing the 3D cultured cells was quickly dried at room temperature in a dark room. An automatic fluorescence microscope (ASTA Scanner^TM^, Medical & Bio Decision, South Korea) was used to acquire the three-color images. The microscope on a moving stage in the scanner automatically focused on the cell spots by moving in the z direction and selecting the highest fluorescent cell image and took 532 individual pictures from a single stained micropillar chip at 4× magnification. The 532 pictures of the cell spots were then consolidated into a single JPEG image for data analysis. From the three-color images, we extracted the area that had a color intensity higher than the threshold intensity to eliminate the background noise.

### 3.5. Viability and Apoptosis Analysis

We used Hoechst 33,342 for Nucleus staining, CellEvent^®^ Caspase-3/7 Green (Thermo Fisher Scientific, Seoul, South Korea) for cell apoptosis, and Calcein AM (4 mM stock from Invitrogen, Seoul, South Korea) for cell viability. Without a complex cell fixing protocol, apoptosis-inducing drugs were identified. Nucleus (Blue) and caspase-3/7 (Green) were stained in the live cell condition, and the cell viabilities were measured by staining the cells with calcein AM (Red). An automatic optical fluorescence scanner (ASFA™ Scanner ST, Medical & Bio Device, Suwon-si, South Korea) was used to measure the red, green, and blue fluorescence intensities using an eight-bit code among the RGB codes (0–255). The area of the 3D cultured cells was identified according to the intensity threshold (20 RGB codes) to reduce the background noise. The area of the red was used for calculating the cell viability while the blue and green areas were used for calculating apoptosis.

In the case of the relative viability, the red color area of the cells exposed to the drug was divided by the control cell area of the no drug. The relative viabilities are based on the healthy cells without drugs. Six alginate spots were used for calculating the average and standard deviation of the relative viability, as shown in Figure 1. In the apoptosis assay, some cells exposed to the drugs showed fluorescence in the green range (350–550 nm) due to the natural fluorescence of the drugs. To solve this problem, unstained controls were prepared for comparison in the chip layout shown in Figure 1. Among the six alginate spots for one drug, three spots were without the caspased-3/7 staining dye, and the other three spots were with the caspase-3/7 staining dye (Figure 1). To measure the activated caspase-3/7, the average cell area of the stained cells on three alginate spots was compared with one of the unstained cells on the other three alginate spots shown in Figure 1. Because of the background noise of the green fluorescence in the stained cells, the *p*-values of the cell areas between the stained and unstained cells were a statistical criterion to measure the activated caspase-3/7. If the *p*-value was higher than 0.05, the stained cells were the same as the unstained cells, and the apoptosis rate was zero regardless of the green cell area. In case of a *p*-value below 0.05, the apoptosis rates were calculated shown in Table 1. The apoptosis rates of one alginate spot were calculated as below:(1)$$\mathrm{Apoptosis}\;\mathrm{rate}\left[\%\right]\;=\;\frac{A_{c1}-A_{c2}}{A_{b}}$$
where A_c1_ is the area of the caspase-3/7 stained cell, and A_b_ is the nucleus area of the cells. A_c2_ is the average area of the Caspase-3/7 unstained cells in three alginate spots. The apoptosis rates were calculated for the three spots stained with Caspase-3/7. Table 1 and Figure 4 and Figure 5 show the average and standard deviation of the apoptosis rates of the 70 drugs.

### 3.6. Statistical Analysis

To determinate active caspase3/7 expression, we used *p*-value. *P*-values were calculated by the green area of the stained and unstained cells with caspase-3/7. The stained and unstained spots were three, respectively. TTEST function in Excel2016 was used. The conditions were two-tailed distribution and paired type. If the *p*-value was higher than 0.05, the two groups were not significantly different and active caspase3/7 did not expressed. Thus, we calculated the apoptosis rate in the drugs with *p*-value lower than 0.05.

## 4. Conclusions

A high throughput apoptosis assay based on a 3D cell culture was developed with a micropillar/microwell chip platform. The micropillar/microwell chip could miniaturize the live apoptosis assay and subsequently reduce the assay volume to under 1 uL for staining caspase-3/7. In addition, drugs with autofluorescence could be distinguished in the live cell apoptosis assay in a high-throughput manner. As a proof of concept, seventy drugs were screened in a chip which included unstained cells for all the drugs to exclude autofluorescence drugs. Four drugs (Bosutinib, PHA-665752, Sunitinib Malate, and Sotrastaurin) from among the seventy drugs showed green autofluorescence in the unstained cells and were excluded from the activated caspase-3/7 analysis. In the activated caspase-3/7 analysis, four drugs (Cediranib, Cabozatinib, Panobinostat, Carfilzomib) showed highly activated caspase-3/7. This means the drugs induced apoptosis and killed the 3D cultured cells. With western blot of caspase-3/7, the chip results were verified.

Declaration of Conflicting Interests: The authors declare no potential conflicts of interest with respect to the research, authorship, and/or publication of this article.

## Figures and Tables

**Figure 1 molecules-24-03362-f001:**
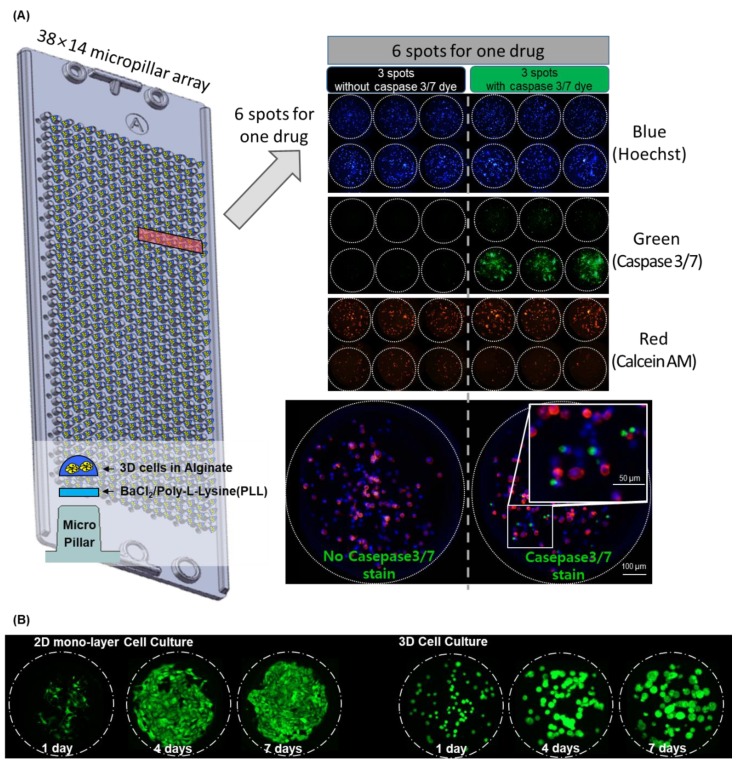
Schematic view of the apoptosis assay chip with live 3D cells. Three-color staining for nucleus (Blue), caspase3/7 (Green), and calcein AM (Red). (**A**) In a single chip (38 × 14 micropillar array), we tested one drug in one block (1 × 6 micropillar array) and 72 drugs (including two controls) in a 36 × 2 block array. Six spots were used for one drug. Three spots were without the caspase-3/7 staining dye, and the other three spots were with the caspase-3/7 staining dye. (**B**) 2D and 3D cells grown in the micropillar chip [13].

**Figure 2 molecules-24-03362-f002:**
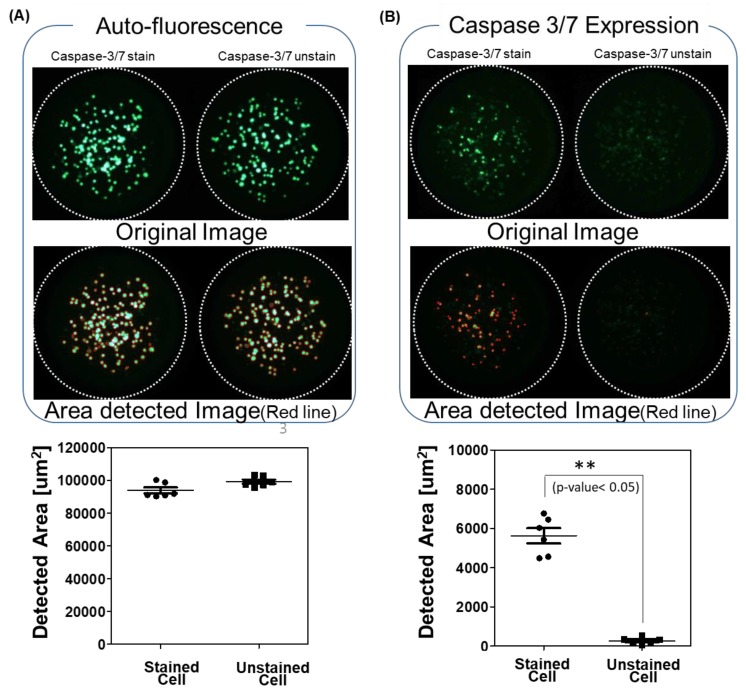
Scanning image of two drugs. (**A**) Sotrastaurin (#51 drug) shows autofluorescence; (**B**) Cediranib (#30 drug) shows activated caspase 3/7. Activated caspase 3/7 is detected when the fluorescence area of the caspase 3/7 dye differs significantly from the no dye condition under a *p*-value of 0.01. Red line indicates the areas with an intensity above 20 Green eight-bit codes (0–255).

**Figure 3 molecules-24-03362-f003:**
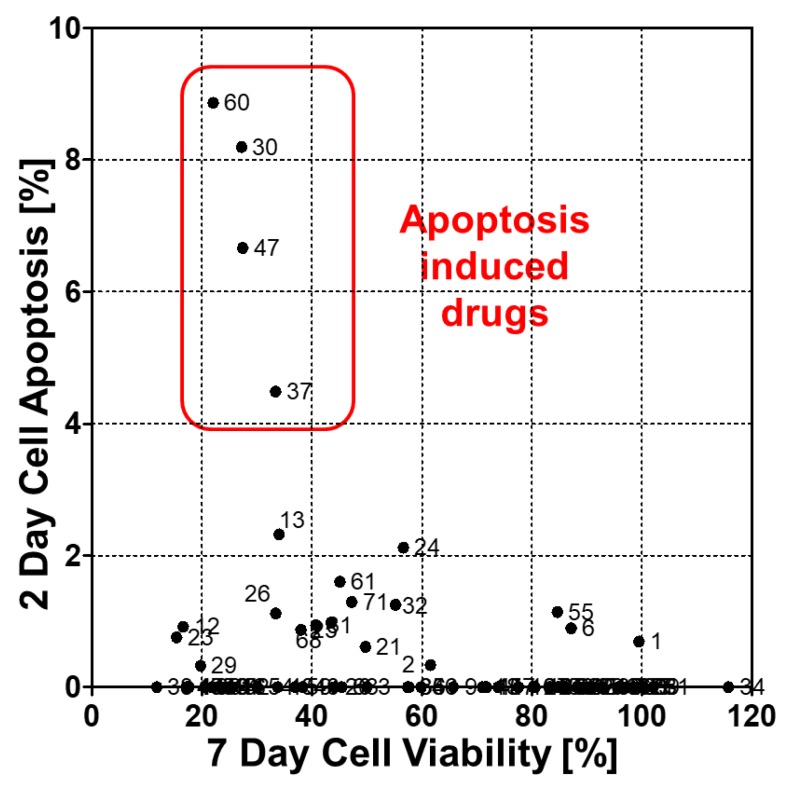
Detection of apoptosis-inducing drugs among the 70 drugs by comparing with activated caspase-3/7 at day 2 and cell viability at day 7. The number around the dot is the drug number in Table 1.

**Figure 4 molecules-24-03362-f004:**
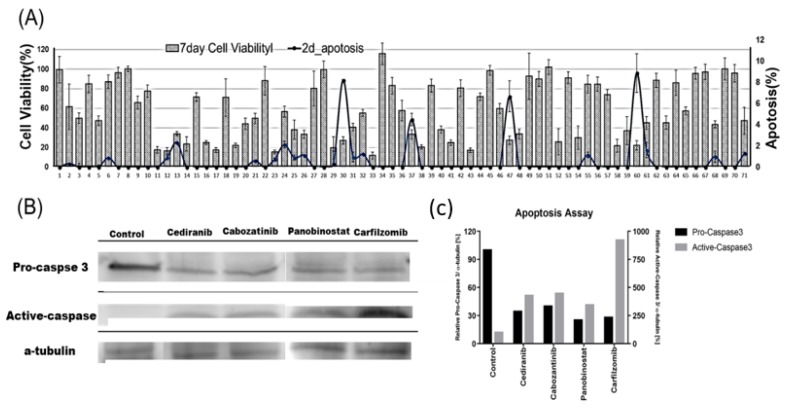
Apoptosis assay results. (**A**) Cell viability and apoptosis of the 70 drugs. (**B**) Eleven highly apoptotic drugs. (**C**) Western blot assay of four apoptotic drugs.

**Figure 5 molecules-24-03362-f005:**
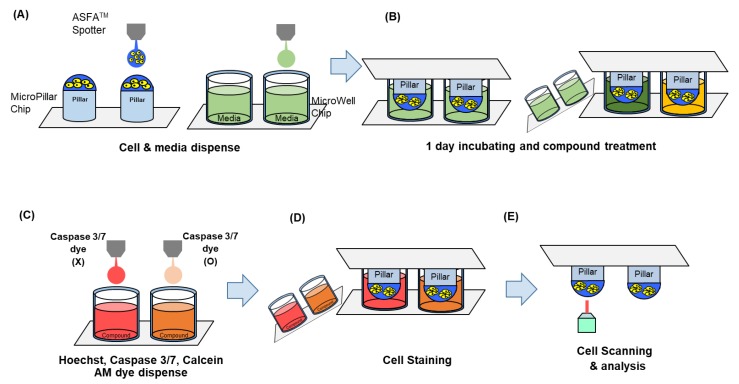
Experimental procedure. (**A**) Cells and media are dispensed on a micropillar and in a microwell, respectively. (**B**) Cells are immobilized in alginate onto the top of the micropillars and dipped in the microwells containing the growth media for one day to stabilize the cells. Then, the microwell chips filled with the drugs are used to replace the old one for the drug treatment. (**C**) Three dyes (Hoechst, Caspase3/7, and Calcenin AM) are dispensed into the microwells. (**D**) Cells are exposed to the dyes by moving the micropillar chip to a new microwell chip. (**E**) Three-color fluorescence images are scanned for the data analysis.

**Table 1 molecules-24-03362-t001:** Relative cell viabilities and apoptosis rate.

Drugs	Apoptosis Rate [%] at Day 2		Cell Viability [%] at Day 7		Drugs	Apoptosis Rate [%] at Day 2		Cell Viability [%] at Day 7	
	Mean	SD	Mean	SD		Mean	SD	Mean	SD
1_DMSO	0.69	0.01	99.42	9.16	37_Cabozantinib	4.49	0.59	33.38	3.01
2_AEE788	0.34	0.10	61.57	4.03	38_Foretinib	0	0	20.45	3.13
3_Afatinib	0	0	49.98	2.47	39_Ibrutinib	0	0	83.26	7.65
4_BMS-599626	0	0	84.89	7.01	40_Vemurafenib	0	0	38.29	1.93
5_Erlotinib HCl	0	0	47.45	4.65	41_Trametinib	0	0	24.93	1.90
6_Dacomitinib	0.89	0.19	87.11	10.32	42_LDE225	0	0	80.57	11.47
7_Gefitinib	0	0	96.42	7.42	43_LDK378	0	0	17.12	1.48
8_Lapatinib	0	0	100.23	2.22	44_LEE011	0	0	71.75	0.32
9_Neratinib	0	0	65.71	0.96	45_Nilotinib	0	0	98.56	3.70
10_CI-1033	0	0	77.57	5.42	46_Olaparib	0	0	59.86	6.60
11_CO-1686	0	0	17.64	2.53	47_Panobinostat	6.66	1.44	27.42	3.96
12_BKM120	0.92	0.35	16.56	2.60	48_Pazopanib HCl	0	0	33.76	5.86
13_BYL719	2.32	0.22	34.03	2.95	49_PD 0332991	0	0	92.83	30.81
14_XL147	0	0	23.57	4.21	50_PF-04449913	0	0	89.99	9.74
15_Everolimus	0	0	71.49	6.86	51_Sotrastaurin	0	0	102.03	8.52
16_AZD2014	0	0	24.98	1.68	52_Sunitinib Malate	0	0	25.59	17.95
17_PF-05212384	0	0	17.38	1.92	53_Tandutinib	0	0	91.14	7.34
18_XL765	0	0	71.00	8.48	54_Tivozanib	0	0	30.03	16.26
19_BEZ235	0	0	22.00	2.73	55_Vismodegib	1.14	0.26	84.62	10.29
20_AZD5363	0	0	44.08	7.03	56_ PHA-665752	0	0	84.60	3.94
21_ABT-199	0.61	0.37	49.72	7.19	57_Dabrafenib	0	0	73.82	7.10
22_ABT-888	0.00	0.00	88.16	16.55	58_Regorafenib	0	0	21.79	9.13
23_AUY922	0.76	0.15	15.38	1.60	59_Bosutinib	0	0	36.95	21.63
24_Axitinib	2.12	0.10	56.63	2.39	60_Carfilzomib	8.86	1.83	22.05	0.61
25_AZD4547	0.87	0.31	38.02	8.25	61_Ruxolitinib	1.60	0.73	45.07	7.13
26_AZD6244	1.12	0.23	33.45	3.32	62_Vandetanib	0	0	88.58	3.39
27_LGK-974	0	0	80.21	2.08	63_TMZ	0	0	45.39	6.55
28_BGJ398	0	0	99.67	5.23	64_Amorolfine	0	0	86.03	5.40
29_Bortezomib	0.33	0.14	19.77	17.33	65_Mevastatin	0	0	57.40	2.01
30_Cediranib	8.19	1.17	27.19	3.64	66_Amiodarone	0	0	95.74	1.28
31_Crizotinib	0.94	0.06	40.77	3.93	67_Fluvastatin Na	0	0	97.19	11.40
32_Dasatinib	1.25	0.46	55.20	5.23	68_Mycophenolic acid	0.98	0.52	43.50	1.93
33_Dovitinib	0	0	11.78	1.08	69_Raloxifene HCl	0	0	100.19	14.60
34_Imatinib	0	0	115.71	8.27	70_Astemizole	0	0	95.88	10.87
35_INCB28060	0	0	83.07	8.93	71_Fenretinide	1.29	0.11	47.24	20.81
36_LY2835219	0	0	57.65	4.79	-	-	-	-	-

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
