# Peer review of "A High Throughput Apoptosis Assay using 3D Cultured Cells"

_molecules, 2019, doi:10.3390/molecules24183362_

Round 1

Reviewer 1 Report

The current manuscript describes the validation of apoptosis assays with 3D cultures in the micropillar/microwell chip platform. 

Major Comments:

In the abstract, the authors state that “previous 3D culture system in 96- or 384-well plate did not enough capability of high throughput apoptosis based on live cells”. This can be misleading. There are assay kits and literature that demonstrate measurement of apoptosis in 3D cultures in a high throughput manner.  It would be better if the authors are more specific in describing which aspects are supposedly suboptimal. I do agree that a strength of their approach is the highly miniaturized format (less reagent usage). However, since the micropillar/microwell chip platform requires arguably more specialized equipment and requires more steps (instead of add and read type assays), it is difficult to assess whether this approach is truly high throughput.  This paper would benefit from further discussion as to where they think the approach fits in the drug discovery process and what other hurdles need to be overcome that will promote broader usage. This manuscript needs to be edited for English spelling and grammar (tense, plural/singular verbs, missing prepositions and punctuations, etc). Some of the suggested edits are listed below. Notably, the would-be editors should have some scientific background to ensure that the word choices and phrases are scientifically sound. For example, “apoptosis induced drugs” in many places throughout the manuscript should be changed to “apoptosis-inducing drugs” to properly describe the results Line 25: “expansive” should be changed to “expensive” Line 28-29:“drugs…induced apoptosis when cells were dead” is erroneous. Dead cells cannot induce apoptosis pathways, as the phrase might suggest. Line 66: “block the cells” is not clear. I think it will suffice to say “eliminating fixing the cells” or “eliminating cell fixation”. Line 86: Please clarify “replaced with old one”. Line 147-149: The description of the analysis is not quite clear to me. What does “use caspase 3/7 expression” mean? The use of the term caspase 3/7 “expression” or “overexpression” in the manuscript is misleading because there are 2 forms (pro and activated) and the regulation of caspase during the induction of apoptosis is post-translational (expression reflects transcriptional control). It would be more scientifically accurate to refer to the caspase assay output as caspase activation, which would in turn be a reflection of active caspase in Westerns. Line 155: How did the authors arrive at 6h incubation time? It seems long for the reagents involved. Was this optimized for 3D culture? Line 215: Shouldn’t this be activated caspase 3 and not pro-caspase form? Line 236: “excused” is unclear. Did you mean excluded or actually distinguished and analyzed differently?

Minor Comments and Edits:

Reference to Fig.1 appears to be missing from the text. Line 21: Change “were” to “have been” (and the subject should be changed to plural form, ie, assays) Line 42: “reason” should be plural form “reasons” Line 49: Change “had” to “was” Line 53: Change “make” to “makes” Line 49: Change “had” to “was” Line 74: Change “could supply to additional space for control test” to “could provide additional capacity for control tests” Line 85: Change “stabilized” to “stabilize” Line 87: Change “set” to “sets” Line 102: Change “DPAI” to “DAPI” Line 130: Change “voltexed” to “vortexed” and “static” to “incubate”? (the latter one is unclear, as currently stated) Line 145: Change “florescence” to “fluorescence” Line 162: Change “JPGE” to “JPEG” Line 213: Change “highly” to “high” Line 214: Change “3d” to “3D” Line 240: Change “execluded” to “excluded” Line 243: Change “Weston” to “Western”

Author Response

I attached reply file.

Reviewer 2 Report

General comments:

This manuscript describes a workflow to conduct a high-throughput apoptosis assay based on cells cultivated in a three-dimensional format. While the technique itself may be of interest to the audience of the journal Molecules, the key comparison with the data obtained by 384-well plate is missing. Moreover, the results and figures are poorly organized, accompanying with broken sentences throughout the manuscript, which make the manuscript difficult to read.

Specific comments:

Key      assumption: According to the description by the authors, the key advantage      of the proposed assay is the ability to examine cells in a high-throughput      manner with reduced amount of reagent, especially compared with the      384-well plate format (page 2-3). There is NO comparison with the 384-well      plate at all. The reviewer is unable to gauge the advantages of the      proposed approach over 384-well plate, except this one sentence stating      that the reagent consumption is much less than that required by the 384-well      plate. There are many unanswered questions, for example: How are the micropillars      fabricated? How are the cells functionalized on the pillars without being      detached during the media change? How the cell morphology/viability differ      in the proposed format compared to the well plate?

Clarity      of Figures:

(1)  Scale bars are needed for Figure 1 and Figure 3

(2)  Figure 1: What are (X) and (O) in the figure referred to?

(3)  Figure 3 (A) and (B): What does the “Red Line” mean?

(4)  Figure 3 (B): Is the p value 0.01 or 0.05?

(5)  Figure 4: What are the numbers around the dots? 2-Day apoptosis rate? 7-Day cell viability?

(6)  Figure 5: the western blot results have been blocked by a few white boxes?

(7)  In the text (page 11-13): Fig.4A and Fig.4B are referred to Figure 3 (A) and (B). Correct? So no description on what exactly Figure 4 would like to show.

Number      of drugs tested: How many drugs have been tested? 70 drugs described in      the abstract and the Table. Figure 3 caption implies 51 + 30 drugs?

Broken      language:

The manuscript shall be polished further to improve the readability and scientific precision. To name a few:

-       Title: “High throughput 3D cell-based apoptosis assay”. What does “3D cell” mean? Cells shall be in 3D format naturally. If this “3D” is referred to the culture format, then the title shall be revised.

-       Page 1, Abstract: “….in 96- or 384-well plate did not enough capability of high…”: missing a verb 

-       Page 1, Abstract: “…per chip and reduced 28  assay volume of a well under 1 μl…”: “under 1 μl” means less than 1 μl, but the propose assay requires 1 μl

-       Page 11: “Auto-fluorescence is serious problem in muti-color analysis in live cell”: shall be “a” serious problem. Further, autofluorescence would be a problem even for “single color” analysis.

Statistical      analysis: What kind of test was conducted? In Table 1, what does SD refer      to? Why a lot of mean/SD are “0”? What do those “0” mean?

Author Response

I attach reply file

Reviewer 3 Report

The paper is a methodology-focused paper, which is suited for a technical note. The english writing style is poor and must definitely be re-written in all the manuscript to make some parts clear, which are extremely difficult to understand. Also, there are several inconsistencies throughout the manuscript, which make it imperceptible. At this point, without further clarifications, the manuscript cannot be accepted for publication. It was not clear how some part of the work was performed, I suggest major revisions to improve and clarify the work performed.

Was the morphology of the cells in the microchips analyzed? Do these acquire a spheroid-like morphology? If so, was the 3D structure of the cell spheroids taken into account for the analysis? Was it a single plan or there was any type of deconvolution? It would be important to include a figure with the morphology of the cells.

The live assay was performed without fixing the cells but the caspase 3/7 staining was performed with fixing? This is not clear, please elucidate.

How many replicates were performed for each condition?

Figure 1: Does the staining with caspase 3/7 correspond to cells with drug treatment or without? It is not clear the difference between the wells and why there is fluorescence signal in ones and not the others (right panel, top and bottom images). There are no scale bars. In the magnified image there is reference to (X) and (O), what does it refer to? It must be described in the legend of the figure.

Figure 2: Was Dapi or hoescht used for nuclei staining? In the methods hoechst is mentioned but in the figure it is “DPAI”.

For the fluorescent drugs, these could not be analyzed by this assay, right? The background signal was removed from the signal, then apoptosis could not be detected? Or was there any optimization of the protocol performed?

In the text it is mentioned figure 4A and 4B but figure 4 does not contain panel A and B. Please verify. Are the 4 drugs highlighted in figure 4 really inducing apoptosis or are these 4 drugs auto-fluorescent? If we go back to the table data it seems these induce apoptosis but this should be clear. Please clarify.

Did the cells proliferate over the 6 days?

Author Response

I attach reply file

Round 2

Reviewer 2 Report

The authors have spent decent effort in revising the manuscript, however, a few key issues remain clear.

(1) The comparison with 384-wells or other existing approaches. It is still unclear whether the obtained results from the proposed approach are reliable.

(2) Statistical test is important. How did the authors determine the p-value? The statistical test is not described in the manuscript.

(3) Track-changes have kept both the old and new versions of figures, which makes the reading quite difficult. 

Author Response

I attach file.
